# Improvement of Longitudinal Performance Uniformity of Hot-Rolled Coils for Cold-Rolled DP980 Steel

**Haijun Li [1], Tianxiang Li [1],\*, Chaofei Li [2], Zhaodong Wang [1] and Guodong Wang [1]**

[1] The State Key Laboratory of Rolling and Automation, Northeastern University, Shenyang 110819, China; lihj@ral.neu.edu.cn (H.L.); zhdwang@mail.neu.edu.cn (Z.W.); wanggd@mail.neu.edu.cn (G.W.)

[2] Avic Jonhon Optronic Technology Co., LTD, Luoyang 471003, China; li1chaofei@163.com

\* Correspondence: ltx112358@gmail.com; Tel.: +86-159-4018-3785

**Abstract:** Cold-rolled DP980 steel is widely used in the automobile industry. Hot-rolled coil is the raw material of cold-rolled DP980 steel, the head and tail parts of which are usually obviously stronger than the body part. The objective of this study is to improve the longitudinal performance uniformity of hot-rolled coils. The material properties of this steel, such as the dynamic continuous cooling transformation, the influence of the cooling mode before coiling, the cooling rate during coil cooling on the microstructure, and mechanical properties of cold-rolled DP980 steel were investigated through thermal simulation experiments and hot rolling experiments. Meanwhile, the temperature field of hot-rolled coil was analyzed using ABAQUS software, which was used to survey the cause of the longitudinal performance fluctuations of hot-rolled coils, combined with an investigation of the aforementioned material properties. The results illustrate that the average cooling rate of the head and tail parts are higher than that of the body part during coil cooling, which causes the longitudinal performance fluctuation of hot-rolled coils. Based on the temperature field of hot-rolled coil, obtained by FEM, the parameters of the U-shaped cooling process were optimized and used in industrial applications.

**Keywords:** DP980 steel; U-shaped cooling; dynamic continuous cooling transformation; microstructure and mechanical properties

## 1. Introduction

As a class of advanced high-strength steels, dual-phase steels, which consist of martensitic islands in a ferrite matrix have excellent mechanical properties, such as a high strength and plasticity, low yield ratio, and high initial work hardening rate. Together, these properties give them a good combined performance, and makes them suitable for automotive applications [1,2]. The longitudinal performance of hot-rolled coil, the raw material of cold-rolled DP980 steel, fluctuates greatly. The strength of the head and tail parts are obviously higher than that of the body part of the 2250 mm hot rolling plant from the Handan iron and steel company of the HBIS group in China. Therefore, it is necessary to cut off the head and tail before cold rolling, which greatly reduces the material yield.

The main factors influencing the strength of micro-alloyed steel include the microstructure and second-phase particle precipitation [3]. During and after coiling, the temperature of the inner layer and outer layer of hot-rolled coil drops relatively quickly due to direct contact with the air or coiler drum [4]. The enhancement effect of the second particle precipitation on the strip head and tail parts after coiling is weakened due to a high cooling rate, which is beneficial for reducing the strength of the head and tail parts of hot-rolled steel coils and improving the uniformity of the longitudinal properties

of hot-rolled coils. Therefore, the influence of second-phase particle precipitation on the longitudinal performance uniformity of hot-rolled coils can be excluded. The difference in the microstructure between the head, tail, and middle part is the key factor that leads to the performance fluctuation of the hot-rolled coils. In hot-rolled steel strips, the microstructure and mechanical properties are greatly affected by process parameters, such as the rolling temperature, cooling mode, cooling rate, and the coiling temperature [5–10]. The variation of the temperature of hot-rolled coil plays an important role in the microstructure and mechanical properties during coil cooling. Hence, it is important to obtain the temperature field of cooling coil [11]. Currently, hot-head and hot-tail treatment technology is usually used to improve the longitudinal performance uniformity of hot-rolled steel coils in industrial production. However, the cooling parameters of this technology are mainly determined based on experience and lack theoretical support.

In this paper, considering the fluctuation of the mechanical properties along the longitudinal direction of hot-rolled coils, the influence of the cooling mode before coiling and the cooling rate during coil cooling on the microstructure and mechanical properties of cold-rolled DP980 steel were studied. Meanwhile, the temperature field of hot-rolled coil was analyzed using ABAQUS software, (ABAQUS 6.14, Dassault Systemes Simulia Corp., Providence, RI, USA) which was used to survey the cause of the longitudinal performance fluctuations of hot-rolled coils, combined with thermal simulation experiments and hot rolling experiments. Based on the temperature field of hot-rolled coil, obtained by finite element method (FEM), the parameters of the U-shaped cooling process were optimized and used in 2250 mm hot strip mill (HSM) from the Handan iron and steel company.

## 2. Experimental Method and FEM Simulation to Obtain the Temperature Field of Hot-Rolled Coil

### 2.1. Material and Experimental Method

The experimental materials were obtained from an industrialized continuous casting slab of cold-rolled DP980 steel. The chemical composition of the casting steel is listed in Table 1. In order to study the dynamic continuous cooling transformation and the influence of the cooling rate on the properties during and after coiling, thermal simulation experiments were conducted on an MMS-300 thermal simulator. Hot rolling experiments were conducted on a $\phi$450 mm × 450 mm two-high reversing hot mill to study the effects of the cooling rate before coiling on the strip properties.

**Table 1.** Chemical composition of the experimental steel, wt%.

| C | Si | Mn | P | S | Nb | Cr | Als |
|---|---|---|---|---|---|---|---|
| 0.09 | 0.55 | 2.4 | 0.015 | 0.01 | 0.04 | 0.5 | 0.049 |

### 2.1.1. Thermal Simulation Experiments

Hot rolling experiments were conducted on a $\phi$450 mm × 450 mm two-high reversing hot mill. The maximum rolling force and roll gat are 4000 kN and 170 mm, respectively. The rated power of main drive motor is 400 kW. The multi-function cooling system is installed on the delivery side of the mill. A slab with 110 mm thickness was reheated to and maintained at 1250 °C in a furnace for 1 h, then rolled to a plate with 12 mm thickness. The plate was cooled to room temperature by air, after rolling. The cylindrical samples, with a size of $\phi$8 mm × 15 mm, were cut from the plate. A schematic diagram of the experiment is presented in Figure 1.

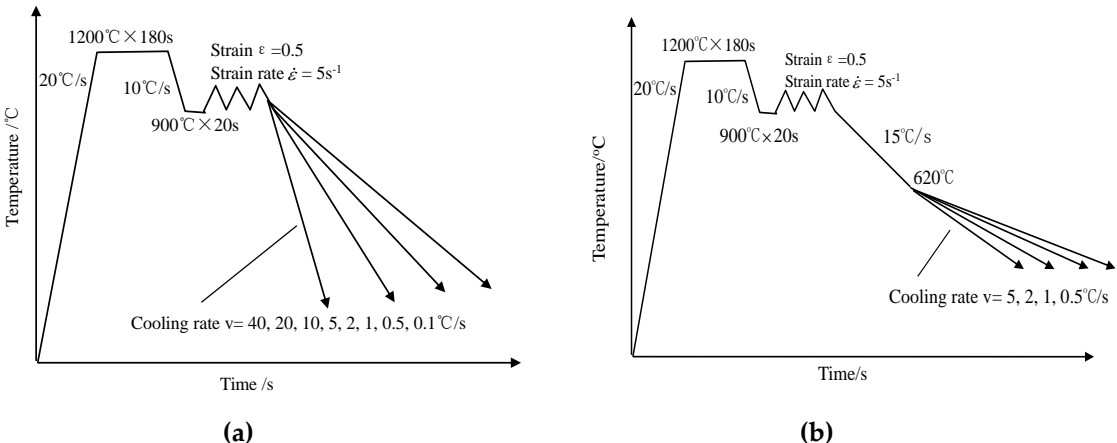

**Figure 1.** Schematic diagram of the thermal simulation experiments. (**a**) Dynamic continuous cooling transformation; (**b**) effects of the cooling rate during coiling and after coiling.

The cylindrical samples were heated to 1200 °C, with a 20 °C/s heating rate, and maintained for 180 s. They were then cooled to 900 °C, with a 10 °C/s cooling rate. The samples were maintained at 900 °C to homogenize the temperature distribution. The samples were compressed with a 0.5 strain, at a strain rate of 0.5 s$^{-1}$. As shown in Figure 1a, after compression deformation, the samples used to study the dynamic continuous cooling transformation were cooled to below 100 °C, at different cooling rates. The change of expansion with temperature was recorded during the experiment. The phase transition point was determined with the tangent method. The curves of expansion with temperature under different cooling rates were recorded during the experiment. When phase transformation occurs, the volume change caused by the phase transformation accumulates on the expansion curve, which destroys the linear relationship between the expansion amount and the temperature. The temperature corresponding to the starting point and the ending point of the change phase of the expansion curve are the phase transformation starting temperature and finishing temperature, respectively, which were determined using the tangent method in this paper. As shown in Figure 1b, after compression deformation, the samples used to study the effects of the cooling rate after coiling on the properties were cooled to 620 °C (coiling temperature) at a cooling rate of 15 °C/s, then cooled to below to 100 °C, at cooling rates of 10 °C/s, 1.0 °C/s, and 0.5 °C/s. The specimens cut from the compressed samples were ground, polished, and then etched in a 4% nitrate alcohol solution. The microstructure was observed using an optical microscope (OLYMPUS BX53MRF, Tokyo, Japan). The Vickers hardness of the samples used to study the effects of the cooling rate during coiling and after coiling on the properties was tested too.

### 2.1.2. Hot Rolling Experiments

The maximum rolling force and roll gat of the hot mill are 4000 kN and 170 mm, respectively. The rated power of the main drive motor is 400 kW. The multi-function cooling system is installed on the delivery side of the mill. The schematic diagram of the hot rolling experiments is presented in Figure 2. The slabs with a 70 mm thickness were reheated to 1200 °C in a furnace and maintained for 2 h. Then, they were hot rolled to sheets with 4 mm thickness by multi-pass rolling. The final rolling temperature was 900 °C. Following rolling, the plates were cooled to 620 °C (industrial coiling temperature) in different cooling modes, such as air-cooling, slow-cooling with water, and fast-cooling with water. Then, the plates were slowly cooled to room temperature in a pit-type furnace. Tensile testing was performed on a normal tensile sheet specimen to study the strength of different cooling processes. The thickness of the tensile specimen was 3.8 mm, and the total length was 200 mm. The lengths of the clamping part and body part were 50 mm and 70 mm, respectively. The clamping part and body part were smoothly connected with an arc with a radius of 20 mm. The width of the body part is

12.5 mm. The specimens cut from the hot-rolled plates were ground, polished, and then etched in a 4% nitrate alcohol solution to study the microstructure with different cooling processes. The optical micrograph (OM) and scanning electron micrograph (SEM) was observed using an optical microscope (OLYMPUS BX53MRF, Tokyo, Japan) and scanning electron microscope (FEI Quanta 600, Hillsborough, OR, USA), respectively.

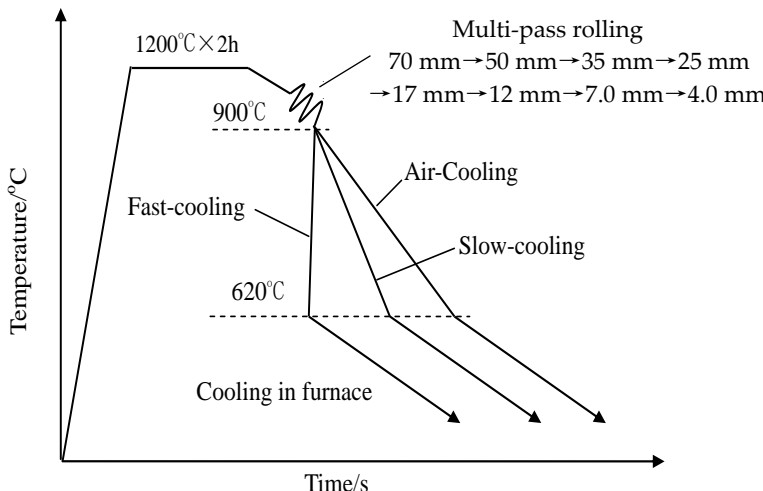

**Figure 2.** The schematic diagram of the hot rolling experiments.

## 2.2. FEM Simulation to Obtain the Temperature Field of Hot-Rolled Coil

The FEM simulation was conducted on a hot strip with a thickness of 2.5 mm and width of 1250 mm using ABAQUS software. The coiling temperature was 620 °C. The diameter of the coiler drum was 760 mm, and the external diameter of the hot-rolled coil was 1860 mm. Taking the axis of the coiler mandrel as the symmetry axis, the hot-rolled coil was meshed by an eight-node linear heat transfer hexahedron element, and the element type was DC3D8. In order to reduce the computation time, the meshes of the middle layers were coarser than those of the outer and inner layers. As shown in Figure 3, the FEM simulation was divided into two stages: The coiling stage and the coil discharged stage. The hot-rolled coil was regarded as a multi-layered cylinder. During the coiling stage, each k layer coiled was treated as a tick step, and the cooling time was the time for coiling the k layer strip. The cooling time of the last step (n = 15) was the time for discharging the coil. The value k for the strip head and tail was smaller than that of the strip body. The value k, thickness, and cooling time of each tick step are listed in Table 2. In each tick step, coils with new diameters were re-modeled. By defining the node data in the input file, the final temperature of the previous tick step was treated as the initial temperature of the old layers of the current tick step, and the coiling temperature of 620 °C was treated as the initial temperature of the new layers of the current tick step.

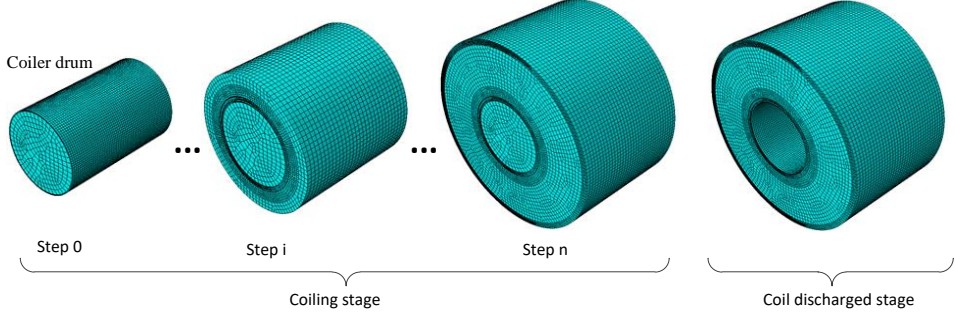

**Figure 3.** The finite element model of hot-rolled coil.

**Table 2.** Tick step information for the FEM simulation of the temperature field of hot-rolled coil during coiling.

| Tick Step | Value k | Step Layer Thickness /mm | External Diameter of Coil/mm | Cooling Time /s |
|---|---|---|---|---|
| 0 | 0 | 0 | 760 | 0 |
| 1 | 4 | 10 | 780 | 0.63 |
| 2 | 4 | 10 | 800 | 0.65 |
| 3 | 4 | 10 | 820 | 0.67 |
| 4 | 8 | 20 | 860 | 1.40 |
| 5 | 20 | 50 | 960 | 3.91 |
| 6 | 40 | 100 | 1160 | 9.64 |
| 7 | 40 | 100 | 1360 | 11.09 |
| 8 | 40 | 100 | 1560 | 12.73 |
| 9 | 20 | 50 | 1660 | 6.78 |
| 10 | 20 | 50 | 1760 | 7.18 |
| 11 | 8 | 20 | 1800 | 2.94 |
| 12 | 4 | 10 | 1820 | 1.49 |
| 13 | 4 | 10 | 1840 | 1.50 |
| 14 | 4 | 10 | 1860 | 1.52 |
| 15 | 0 | 0 | 1860 | 43.00 |

A hot-rolled coil can be considered as a periodically laminated material of a steel layer and interface layer in the radial direction [12]. The axial and circumferential directions are parallel to the contact interface between steel layers. In those directions, the equivalent thermal conductivity is almost the same as the thermal conductivity of steel. However, the equivalent thermal conductivity may be quite different from that of steel. The contact interface of steel layers consists of actual contact spots between the asperities on each surface, and voids among the actual contact spots. The thermal resistance of the contact interface is composed of the thermal resistance of the actual contact spots, and the thermal resistance of the cavity in parallel. The equivalent thermal resistance of the hot-rolled coil in the radial direction is composed of the thermal resistance of the contact interface, and the thermal resistance of a steel layer in a series. The equivalent thermal conductivity of hot-rolled coil in the radial direction can be calculated with the following equation [13]:

$$\lambda_d = \frac{(4P_c\lambda\delta + c\pi R_i\sigma_b\lambda_v)\lambda h}{4P_c\lambda\delta h + c\pi R_i\sigma_b\lambda_v h + 2\lambda c\pi R_i\sigma_b\delta} \tag{1}$$

where $\lambda_d$ is the equivalent thermal conductivity of hot-rolled coil in the radial direction, W/(m·K); $\lambda$ and $\lambda_v$ are the thermal conductivity of steel and voids, respectively, W/(m·K); $P_c$ is the nominal compressive stress, MPa; $\delta$ is the roughness of the steel surface, μm; $c$ is the contact coefficient of the interface, which is about 2.5–3.0; $R_i$ is the radius of the contact spot, μm; $\sigma_b$ is the yield stress of steel, MPa; and $h$ is the strip thickness, mm. For a strip with a thickness of 2.5 mm, when $P_c$ is equal to 30 MPa and $R_i$ is equal to 0.4 μm, the ratio of the thermal conductivity of steel to the equivalent thermal conductivity of hot-rolled coil in the radial direction, $\lambda/\lambda_d$, is almost equal to 6.0. The value of the thermal conductivity of steel, $\lambda$, is listed in Table 3 [12].

**Table 3.** Tick step information for the FEM simulation of the temperature field of hot-rolled coil during coiling.

| Temperature/°C | 100 | 200 | 300 | 400 | 500 | 600 | 700 |
|---|---|---|---|---|---|---|---|
| $\lambda$/W·m$^{-1}$·K$^{-1}$ | 57.8 | 53.2 | 49.4 | 45.6 | 41.0 | 36.8 | 33.1 |

The initial temperature of the coiler drum is assumed to be 50 °C, and the environment temperature is assumed to be 20 °C. The difference of the strip temperature, after laminar cooling along the transverse

direction, is ignored. The initial temperature of the strip, before coiling, is set to 620 °C (coiling temperature). The heat transfer mechanism of the inner layer of hot-rolled coil between the coiler drum is heat conduction during the coiling stage. The heat transfer mechanism of the inner layer surface of hot-rolled coil is heat conduction during the coil discharged stage. The heat transfer mechanism of the outer layer surface and side of the hot-rolled coil includes heat radiation and air convection, and the heat loss due to radiation is much higher than that due to air convection. Radiation heat losses from the strip to the environment can be calculated using the Stefan–Boltzmann equation [14]:

$$\alpha_a = \varepsilon \times \sigma \left[ (T_s + 273)^2 + (T_a + 273)^2 \right] \times \left[ (T_s + 273) + (T_a + 273) \right] \tag{2}$$

where $T$ is the strip surface equivalent heat transfer coefficient of air cooling, W/(m$^2$·K); $\varepsilon$ is the emissivity of the strip, defined as 0.8 [12]; $\sigma$ is the Stefan-Boltzmann constant, $5.69 \times 10^{-8}$ W/(m$^2$·K$^4$); and $T_s$ and $T_a$ are the temperature of the strip surface and environment, respectively, °C.

## 3. Results and Discussion

### 3.1. Dynamic Continuous Cooling Transformation of DP980

The dynamic Continuous Cooling Transformation (CCT) curve of the DP980 steel and microstructures, at different cooling rates, are shown in the Figures 4 and 5, respectively. When the cooling rate is within the range of 0.1–5.0 °C/s, ferrite and bainite can be obtained, and the fraction of ferrite decreases with an increase in cooling rate. When the cooling rate is within the range of 5.0–40 °C/s, only bainite can be obtained. When the cooling rate is within the range of 1.0–40 °C/s, the phase transition starts at 470–580 °C, which is lower than the target coiling temperature of DP980 during the hot rolling process. Therefore, for DP980 steel, the phase transition occurs after coiling.

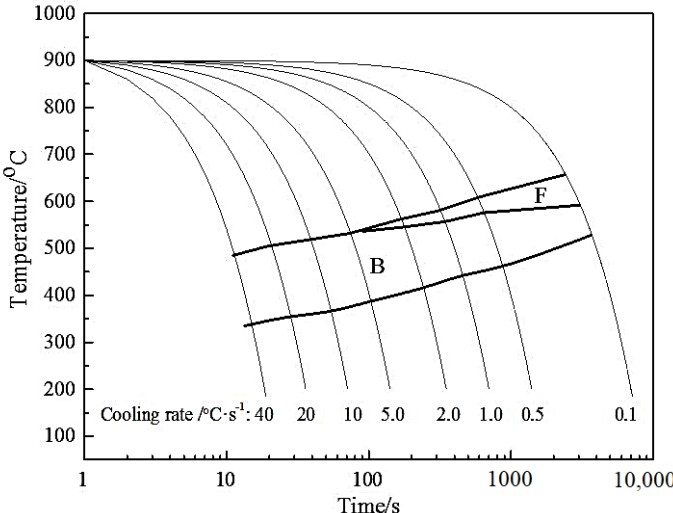

**Figure 4.** Dynamic Continuous Cooling Transformation (CCT) curve of experimental steel.

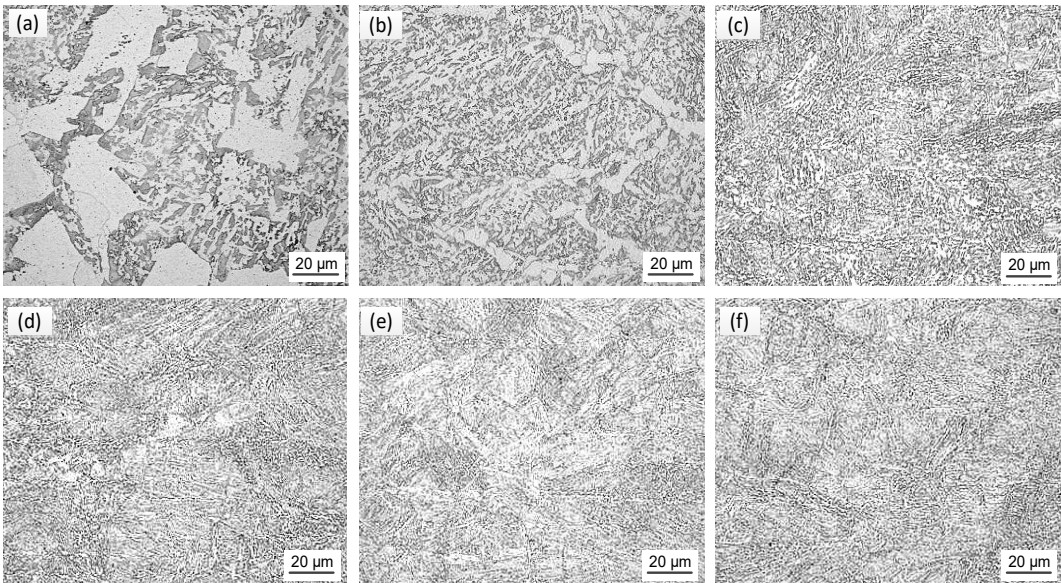

**Figure 5.** Microstructures at different cooling rates: (**a**) 0.1 °C/s; (**b**) 1.0 °C/s; (**c**) 5.0 °C/s; (**d**) 10 °C/s; (**e**) 20 °C/s; and (**f**) 40 °C/s.

*3.2. Influence of the Cooling Rate before Coiling on the Microstructure and Properties of the Strip*

In the hot rolling process of DP980 steel, accelerated rolling is usually used to ensure the final rolling temperature in the industry. The cooling rate of the strip body is usually higher than that of the strip head and tail, because the velocity of the strip body is usually higher than that of the strip head and tail on the run-out table. Hot rolling experiments were conducted to study the effects of the cooling rate before coiling, on the microstructure and properties of the strip. The hot-rolled plates were cooled to a coiling temperature of 620 °C with air-cooling, slow-cooling, and fast-cooling modes, and then slowly cooled to room temperature in a pit-type furnace.

The OM and SEM micrographs of hot-rolled plates, with different cooling modes, are presented in Figure 6. It can be seen, from the OM micrographs (Figure 6a–c), that the microstructure of the experimental steel mainly consists of irregular polygonal ferrite, and a small amount of granular bainite is distributed at the grain boundary of the ferrite. There was no significant difference in the microstructure between the hot-rolled plates with different cooling modes. The size of the ferrite and martensite/austenite (M/A) island become slightly finer with the increase in the cooling rate.

According to the Hall-Petch relationship, the finer grains contribute to improving the strength of materials. The tensile strength and elongation of hot-rolled plates are listed in Table 4. It can be seen that both the tensile strength and elongation of hot-rolled plates are slightly higher with the fast-cooling mode than with the air-cooling or slow-cooling modes. Therefore, the velocity acceleration when the rolling is complete contributes to the increase in the strength of the body and to the improvement of the longitudinal performance uniformity of hot-rolled coil.

**Table 4.** The tensile strength and elongation of experimental steels.

| Number | Cooling Mode | Tensile Strength/MPa | Elongation/% |
|--------|--------------|----------------------|--------------|
| 1 | Air-cooling | 778 | 15 |
| 2 | Slow-cooling | 797 | 16 |
| 3 | Fast-cooling | 801 | 17 |

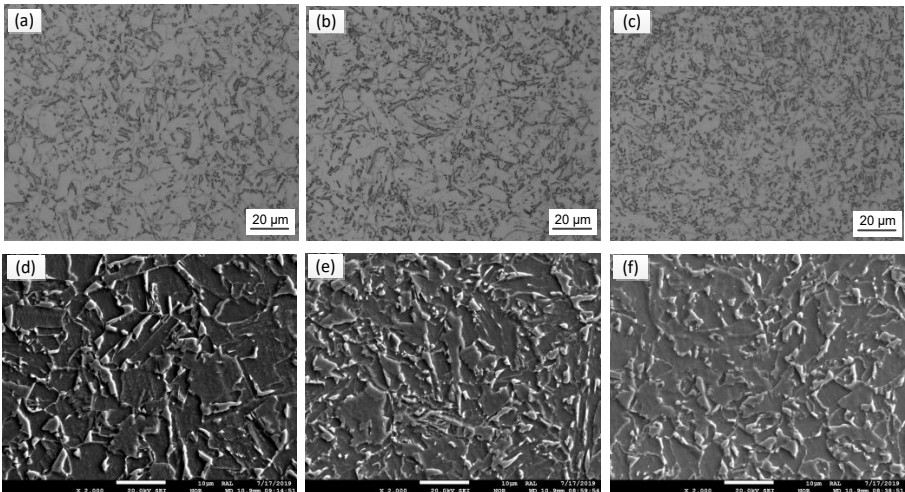

**Figure 6.** Optical micrograph (OM) and scanning electron micrograph (SEM). (**a,d**) Air-cooling; (**b,e**) Slow-cooling; (**c,f**) Fast-cooling.

### 3.3. Influence of the Cooling Rate during and after Coiling on the Microstructure and Properties of Hot-Rolled Coil

Thermal simulation experiments were carried out to study the effects of the cooling rate, during and after coiling, on the microstructure and properties of the strip. After compression deformation, the samples used to study the effects of the cooling rate after coiling, on the properties, were cooled to a coiling temperature at a cooling rate of 15 °C/s, and then cooled to below to 100 °C at different cooling rates. The OM micrographs of the samples with different cooling rates, after coiling, are presented in Figure 7. It can be seen that there are significant differences in the microstructure of experimental steel under different cooling rates. The microstructure mainly consists of proeutectoid ferrite and granular bainite, and the fraction of proeutectoid ferrite decreases obviously with the increase in the cooling rate.

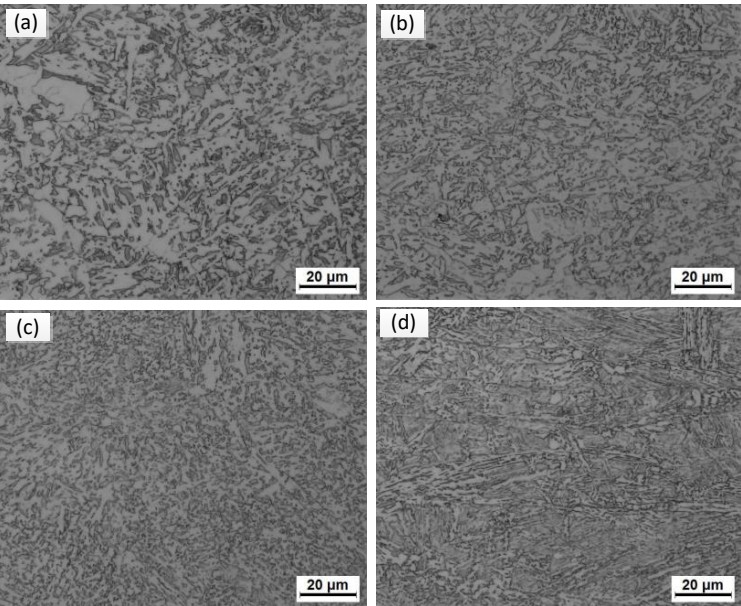

**Figure 7.** Optical micrograph (OM) of samples with different cooling rates, after coiling: (**a**) 0.5 °C/s; (**b**) 1 °C/s; (**c**) 2 °C/s; (**d**) 5 °C/s.

The hardness of experimental steels under different cooling conditions were measured and converted into tensile strength, as listed in Table 5. With the increase in the cooling rate, the hardness

and tensile strength of the experimental steel increased significantly. This is because, with the increment of the cooling rate, the ferrite content reduces and the bainite content increases. The existence of the bainite, as a hard phase, can significantly increase the hardness and strength of the steel. On the other hand, with the increase in cooling rate, the grains of the experimental steel are refined so the effective grain boundary area is increased, which plays an important role in the improvement of the hardness and strength.

**Table 5.** Hardness of experimental steel with different cooling rates.

| Cooling Rate/(°C/s) | 0.5 | 1 | 2 | 5 |
|---|---|---|---|---|
| Hardness/HV | 275 | 280 | 295 | 305 |
| Tensile strength/MPa | 895 | 910 | 954 | 982 |

### 3.4. Analysis of the Temperature Field and Longitudinal Strength Fluctuations of Hot-Rolled Coil

The temperature at the centerline of the inner and outer layers of hot-rolled coil was measured within 6 h after coiler discharging, and the FEM model was verified on the basis of this. The error between the measured and simulated temperatures is within 16 °C. The temperature field of hot-rolled coil at the time of coiler discharging and air cooling for 1 h after coiler discharging are shown in Figure 8. The inner layer of the coil was in contact with the coiler drum during coiling, which caused the inner layer temperature to drop rapidly. When coiler discharging was performed, the inner layer temperature was about 476 °C. However, the temperature of the outer and middle layers of the hot-rolled coil remained at the coiling temperature. After 1 h of coiler discharging, the inner layer temperature increased from 476 °C to 553 °C. The temperature of the outer layer decreased from a coiling temperature of 620 °C to 551 °C. The temperature of the middle layer decreased slightly from a coiling temperature of 620 °C to 609 °C. The cooling paths of the inner, middle, and outer layers of hot-rolled coil, during and after coiling, are presented in Figure 9. It was seen that the cooling paths of the different layers of hot-rolled coil are obviously different. The average cooling rate of the inner and outer layers are higher than that of the middle layers. The strength inhomogeneity of hot-rolled microalloyed steel is mainly affected by microstructure and precipitation of the second phase particle. In the works of Pan [15], the performance uniformity of 700 MPa grade Ti-microalloying high strength steel was researched. The high cooling rate of the inner and outer layers of hot-rolled coil inhibits the precipitation of Ti(C,N), which causes the strengths of the head and tail part of the strip to be lower than that of the body part. However, for DP980 steel, the high cooling rate of the inner and outer layers of hot-rolled coil lead to the decrease in the fraction of proeutectoid ferrite and grains refinement. As a result, the strengths of the head and tail parts were higher than that of the body part of the hot-rolled strip, which is consistent with the problem of industrial production.

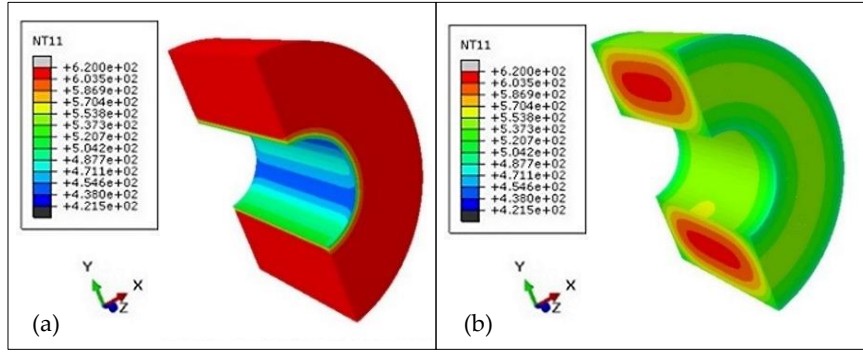

**Figure 8.** Temperature field of hot-rolled coil at different times. (**a**) Coil discharging completed; (**b**) 1 h after coil discharging.

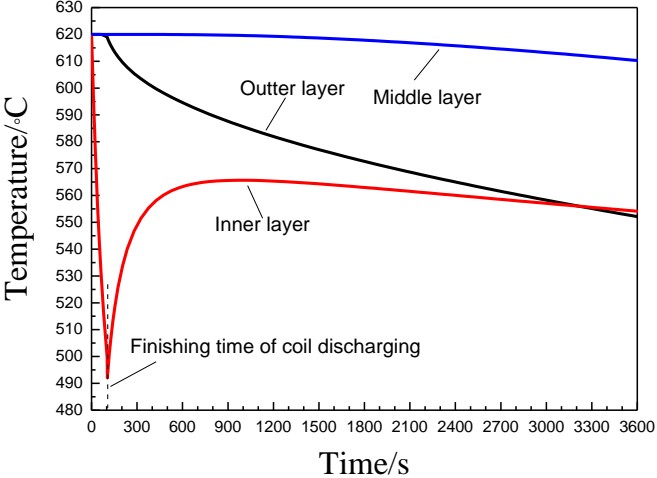

**Figure 9.** The cooling path of the inner, middle, and outer layers of hot-rolled coil, during and after coiling.

## 4. Development of the U-Shaped Cooling Process and Industrial Field Application

As mentioned above, reducing the cooling rate before and after coiling can reduce the strength of the experimental steel. Therefore, the U-shaped cooling process is proposed in this paper to improve the longitudinal performance uniformity of hot-rolled coil, in which the target coiling temperatures of the head and tail parts are increased and are higher than that of the body part. The cooling parameters of the head and tail parts were determined based on the simulated temperature field of hot-rolled coil. The temperature-time curves in different positions of the inner and outer layers are shown in Figures 10 and 11, respectively.

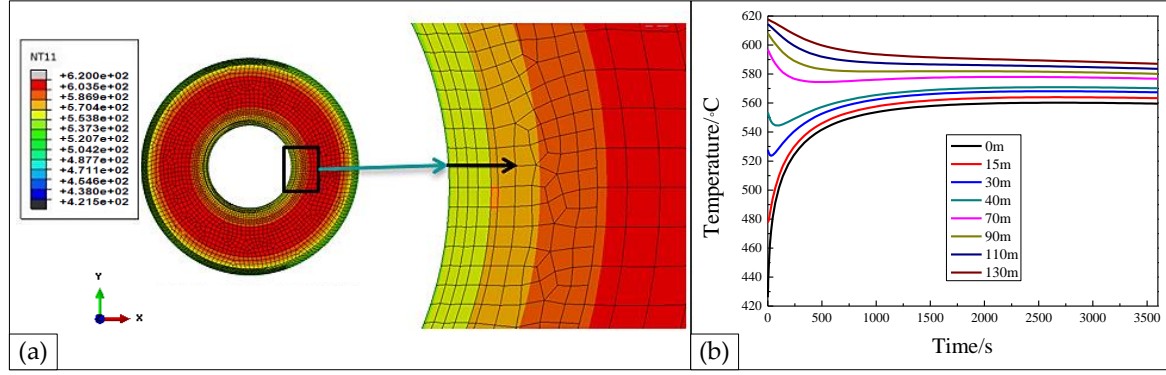

**Figure 10.** Temperature-time curve in different positions of the inner layers: (**a**) temperature field; (**b**) temperature evolution.

In Figure 10, it can be seen that the temperature of the inner layer drops rapidly due to contact with the coiler drum. Especially for the part within 15 m from the strip head, the temperature drops to below 500 °C rapidly from the coiling temperature, when discharging is completed. As the distance from the strip head increases, the trend in the temperature drop slows down. For the part where the distance from the strip heads beyond 70 m, the temperature is not affected by the contact with the coiler drum. Therefore, the cooling parameters of the head part, $\Delta T_H$, $L_{H1}$, and $L_{H2}$, are set to 100 °C, 15 m, and 55 m, respectively, as presented in Figure 12 and Table 6.

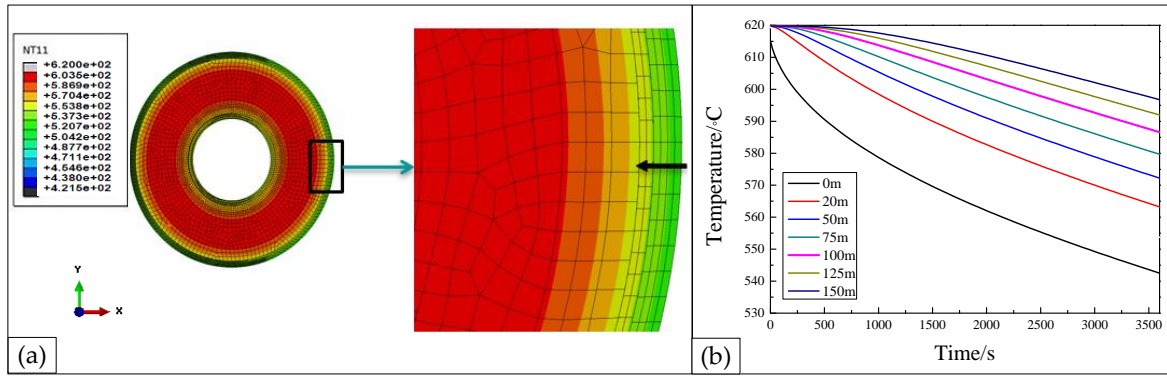

**Figure 11.** Temperature-time curve in different positions of the outer layers: (**a**) temperature field; (**b**) temperature evolution.

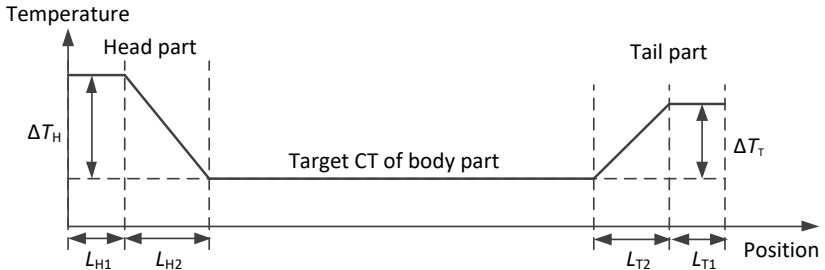

**Figure 12.** The diagram of the U-shaped cooling process.

**Table 6.** Cooling parameters of the head and tail parts of the U-shaped cooling process.

| Head Part | | | Tail Part | | |
|---|---|---|---|---|---|
| $\Delta T_H$/°C | $L_{H1}$/m | $L_{H2}$/m | $\Delta T_H$/°C | $L_{H1}$/m | $L_{H2}$/m |
| 100 | 15 | 55 | 50 | 20 | 80 |

In Figure 11, it can be seen that the temperature of the outer layers decreases with the cooling time increment, due to thermal radiation and air convection. The temperature difference between the part within 20 m from the strip tail end and the strip body part is below 50 °C 1 h after coiler discharging. However, the temperature difference between the part beyond 100 m from the strip tail end and the strip body part is no more than 20 °C 1 h after coiler discharging. Therefore, the cooling parameters of the tail part, $\Delta T_T$, $L_{T1}$, and $L_{T2}$, are set to 50 °C, 20 m, and 80 m, respectively, as presented in Figure 12 and Table 6.

By modifying the head and tail cooling parameters stored in the model database, the industrial application of the U-shaped cooling strategy, discussed above, was realized in 2250 mm hot strip mills of HBIS HANSTEEL in China. The actual coiling temperature curve of hot-rolled coil for cold-rolled steel DP980 is presented in Figure 13. It was seen that the coiling temperature control system accomplished the U-shaped cooling process well.

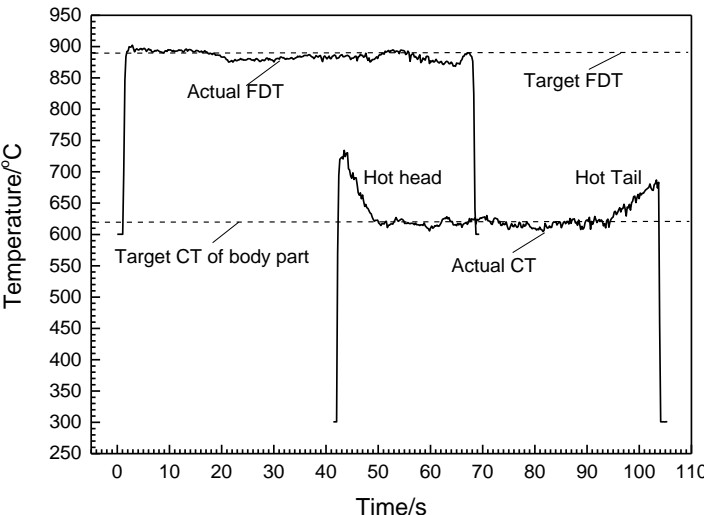

**Figure 13.** The actual temperature curve of hot-rolled coil for cold-rolled steel DP980.

The microstructure of the rolled strips, before and after the U-shaped cooling process application, is shown in Figure 14. The microstructure mainly consisted of proeutectoid ferrite and granular bainite. Before the U-shaped cooling process application, the fractions of the pre-eutectoid ferrite of the head and tail parts were obviously smaller than those of the body part, and the grain sizes of the head and tail parts were also smaller than those of the body part, which caused the head and tail parts to be significantly stronger than the body part, as shown in Figure 15. After the U-shaped cooling process application, similar microstructures were obtained by the head, tail, and body parts. As a result, the longitudinal performance uniformity of hot-rolled coil was obviously improved, as can be seen in Figure 15.

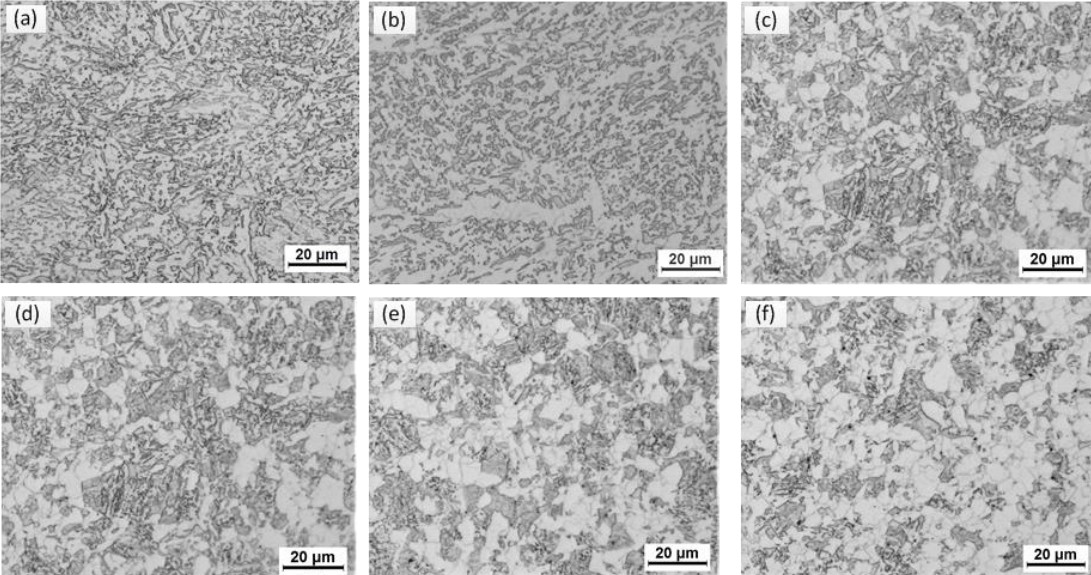

**Figure 14.** Microstructure of the rolled strips, before and after the U-shaped cooling process application: (**a**–**c**) Before the U-shaped cooling process application; (**d**–**f**) after the U-shaped cooling process application; (**a**,**d**) head part; (**b**,**e**) tail part; (**c**,**f**) body part.

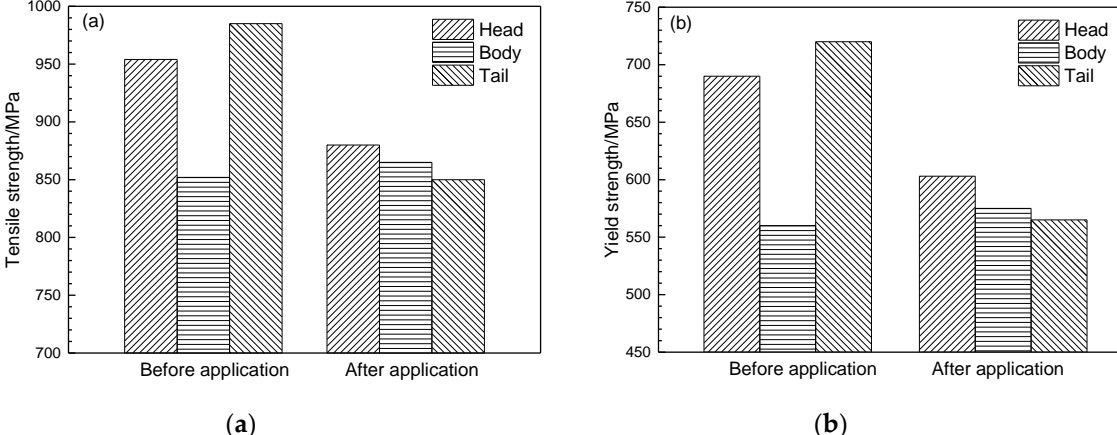

**Figure 15.** The mechanical properties of hot-rolled coils, before and after the application of U-shaped cooling. (**a**) Comparison of the tensile strength; (**b**) comparison of the yield strength.

The lengths cut off from the head and tail parts of hot-rolled coils, before cold rolling, for cold-rolled DP980 steel were both reduced from over 60 m to less than 5 m after U-shaped cooling process application in the 2250 mm hot plant of the Handan iron and steel company of the HBIS group in China.

## 5. Conclusions

The cooling rate, before coiling, had a slight influence on the microstructure and properties of hot-rolled strips. The size of ferrite and the M/A island became slightly finer with the increase in the cooling rate, before coiling. Therefore, the accelerated rolling will contribute to the increase in the strength of the body, and to the improvement of the longitudinal performance uniformity of hot-rolled coils.

The cooling rate, during and after coiling, had an obvious influence on the microstructure and properties of hot-rolled coils, which plays an important role in the longitudinal performance uniformity of hot-rolled coils.

The FEM simulation results showed that the average cooling rate of the inner and outer layers was higher than that of the middle layers of the hot-rolled coil, during and after coiling. For DP980 steel, the high cooling rate lead to the decrease of the fraction of proeutectoid ferrite and grains refinement. As a result, the strengths of the head and tail parts were higher than that of the body part of the hot-rolled strip, which is consistent with the problem of industrial production.

The U-shaped cooling process reduced the cooling rate of the head and tail parts, before and after coiling. Therefore, it can reduce the strength of the head and tail parts. The industrial field application results show that the longitudinal performance uniformity of hot-rolled coil was obviously improved by the U-shaped cooling process.

**Author Contributions:** H.L. designed and wrote the manuscript; T.L. completed the numerical simulation; C.L. helped in the experimental part; Z.W. and G.W. arranged the funding and revised the manuscript. All authors have read and agreed to the published version of the manuscript.

**Funding:** This research was funded by the Fundamental Research Funds for the Central University of China (N170703010).

**Acknowledgments:** We sincerely thank the technicians, who came from the technology center and 2250 mm hot plant of Handan iron and steel company of HBIS group in China, for their assistance.

**Conflicts of Interest:** The authors declare no conflict of interest.

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
