# Peer review of "Improvement of Longitudinal Performance Uniformity of Hot-Rolled Coils for Cold-Rolled DP980 Steel"

_metals, doi:10.3390/met10030382_

Round 1
Reviewer 1 Report
The manuscript describes processes taking place during coiling and the influence of temperature and cooling rate during coiling and after coiling. Also, the problems in the head, body and tail of the strips are investigated. The experimental program was done on DP980 steel. The topic is very actual. According to results from experimental parts and FEM simulation, the U-shaped cooling process for industrial application was designed and tested with very interesting results.
There are typos in the article. And also the part about the experimental methods or characterization is missing. Only 14 references were used in the article. Also, the discussion is missing in the manuscript.
Some information needs to be added for the article to be accepted.
I have the following comments and recommendation for the authors:
- I recommend moving the information on lines 69 through 71 to the previous chapter, where the used material is described. Chapter 2. 1. 1. focuses on the description of thermal simulation experiments.
- The information about MMS-300 simulator is missing.
- Also the information about used microscopes (optical and scanning electron), rolling mill, sample geometry for the tensile test, …
- Line 72, 73, 91: Error in diameter symbol. Instead, the fi symbol is shown.
- How was the temperature for compression deformation selected? And also
- Line 77: mistake in 900 °C
- Please add more information about the dynamic continuous cooling measurement. How were the phase transformation temperatures estimated? Why was the thermal simulator instead of dilatometer used?
- The quality of the images in figure 5 is very low, and the details are not visible in the structure. Please improve.
- How was the presence of retained austenite confirmed? Was the fraction of RA measured?
- The quality of the images in figure 7 and 14 is very low. Please improve.
- Why was hardness measured only in one part of the experiment? It would be interesting to compare the results from the simulator and hot rolling.
- Try to add at least a short discussion of the results with other authors. Although, this is difficult due to the use in the real application.
Author Response
Dear Editors and Reviewers:
Thank you for your letter and comments concerning our manuscript entitled “Improvement of longitudinal performance uniformity of hot-rolled coils for cold-rolled DP980 steel” (ID: metals-740303). Those comments are all valuable and very helpful for revising and improving our paper, as well as the important guiding significance to our researches. We have studied comments carefully and have made correction which we hope meet with approval. We would like also to thank you for allowing us to resubmit a revised copy of the manuscript. In addition, the third author of this paper has changed his work unit after graduation recently, so the work unit information of the third author was updated. The main corrections in the paper and the responds to the reviewer’s comments are as flowing:
Responds to the reviewer’s comments:
Reviewer #1:
- I recommend moving the information on lines 69 through 71 to the previous chapter, where the used material is described. Chapter 2. 1. 1. focuses on the description of thermal simulation experiments.
Response:
Line 68-70, the sentence of “In order to study the dynamic continuous cooling transformation and the influence of the cooling rate on the properties during or after coiling, thermal simulation experiments were conducted on an MMS-300 thermal simulator.” was moved to chapter 2.1
Line 96-96, the sentence of “Hot rolling experiments were conducted on a Ф450 mm × 450 mm two-high reversing hot mill to study the effects of the cooling rate before coiling on the strip properties.” was moved to chapter 2.1 too.
- The information about MMS-300 simulator is missing.
Response:
The MMS-300 thermal simulator is one of the MMS-X00 series thermal simulators, which was developed by Chinese Northeastern University. The MMS-300 thermal simulator have similar functions to Gleeble thermal simulator, and “MMS-300” is the series number information of the thermal simulator.
MMS-300 thermal simulator
- Also the information about used microscopes (optical and scanning electron), rolling mill, sample geometry for the tensile test, …
Response:
1)The following information of rolling mill was added to Chapter 2.1.1.
“The maximum rolling force and roll gat of hot mill are 4000kN and 170mm respectively. The rated power of main drive motor is 400kW. The multi-function cooling system is installed at the delivery side of the mill.”
2) The information of optical microscope was supplied in Chapter 2.1.1
3) The following information of scanning electron microscope and sample geometry for the tensile test was added to 2.1.2.
“The thickness of the tensile specimen is 3.8 mm, and the total length is 200mm. The lengths of clamping part and body part are 50mm and 70mm, respectively. The clamping part and body part are smoothly connected with an arc with radius of 20 mm. The width of body part is 12.5 mm.”
“The optical micrograph (OM) and scanning electron micrograph (SEM) was observed by an optical microscope (OLYMPUS BX53MRF) and scanning electron microscope (FEI Quanta 600), respectively. ”
- Line 72, 73, 91: Error in diameter symbol. Instead, the fi symbol is shown.
Response:
The diameter symbol “φ” in Line 72,73, 91, has been replaced with “Ф”.
- How was the temperature for compression deformation selected? And also
- Line 77: mistake in 900 °C
Response:
The finishing rolling temperature is about 900 °C in industrial production of hot rolled DP980, so the temperature 900 °C was selected as the temperature for compression deformation.
In Line 77, the symbol “900 oC” has been replaced with “900 °C”
- Please add more information about the dynamic continuous cooling measurement. How were the phase transformation temperatures estimated? Why was the thermal simulator instead of dilatometer used?
Response:
1)In order to explain how were the phase transformation temperature estimated, the content of “The change of expansion with temperature was recorded during the experiment. The phase transition point was determined with the tangent method. ” in Line 90~92 was replaced with following content.
“The curves of expansion with temperature under different cooling rate were recorded during the experiment. When phase transformation occurs, the volume change caused by the phase transformation is accumulates on the expansion curve, which destroys the linear relationship between the expansion amount and the temperature. The temperature corresponding to the starting point and the ending point of the change phase of expansion curve are the phase transformation starting temperature and finishing temperature respectively, which were determined with the tangent method in this paper. ”
2)The principle and method of estimating the phase transformation temperatures of MMS-300 thermal simulator are the same as that of the dilatometer, and which will be used depends on how busy the device.
- The quality of the images in figure 5 is very low, and the details are not visible in the structure. Please improve.
Response:
The images in figure 5 and figure 6 were improved.
- How was the presence of retained austenite confirmed? Was the fraction of RA measured?
The samples should be corroded with the Lepera reagent to measure the fraction of RA and confirm the presence of retained austenite. But due to the effect of cooling rate before coiling on the microstructure and properties of hot-rolled coil is non-obvious, this work was not carried out, and the presence of retained austenite was estimated from SEM micrograph. The following content about M-A islands was deleted from this paper.
“It can be seen, from the SEM micrographs (Figure 6 d, e and f), that the M-A islands with short bar-type or irregular polygons are distributed on the coarse polygonal ferrite matrix, and their size is about 2-10 μm.”
- The quality of the images in figure 7 and 14 is very low. Please improve.
Response:
The images in figure 7 and figure 14 were improved.
- Why was hardness measured only in one part of the experiment? It would be interesting to compare the results from the simulator and hot rolling.
Response:
The focus of this paper is the strength uniformity of DP980 steel. According to the thermal compression experiment, limited by the size of the cylindrical samples, the strength of samples cannot be measured directly, so the hardness was measured and converted to strength.
- Try to add at least a short discussion of the results with other authors. Although, this is difficult due to the use in the real application
Response:
The following discussion of the results with PAN’s works was added to Line 267~272.
“The strength inhomogeneity of hot rolled microalloyed steel is mainly affected by microstructure and precipitation of second phase particle. In the works of PAN[15], the performance uniformity of 700 MPa grade Ti- microalloying high strength steel was researched. The high cooling rate of the inner and outer layer of hot-rolled coil inhibits the precipitation of Ti(C,N), which caused the strengths of the head and tail part of the strip are lower than that of the body part. But for DP980 steel, the high cooling rate of the inner and outer layer of hot-rolled coil leads to the decrease of the fraction of proeutectoid ferrite and grains refinement.”

Reviewer 2 Report
The paper presents the results of research on the rolling process of semi-finished products as they are hot-rolled coil of cold-rolled DP980 steel. he authors indicate that in traditional rolling and heat treatment processes the rolling sheets exhibit structural heterogeneity and mechanical properties. The authors point out that thanks to the controlled cooling process and the change of heat treatment parameters, the structure and mechanical properties of rolled sheets can be unified. The work is an interesting study on the impact of rolling process, cooling and heat treatment parameters on sheet properties of DP980 steel. However, I believe that certain things should be supplemented:
- I propose to extend the description of the rolling process by the diagram, parameters in subsequent culverts, order of operations.
- I propose to explain in detail how subsequent layers of sheet metal were modeled in hot-rolled coil.
- Has the sheet strength test been carried out? If so, propose to include the results.
Author Response
Dear Editors and Reviewers:
Thank you for your letter and comments concerning our manuscript entitled “Improvement of longitudinal performance uniformity of hot-rolled coils for cold-rolled DP980 steel” (ID: metals-740303). Those comments are all valuable and very helpful for revising and improving our paper, as well as the important guiding significance to our researches. We have studied comments carefully and have made correction which we hope meet with approval. We would like also to thank you for allowing us to resubmit a revised copy of the manuscript. In addition, the third author of this paper has changed his work unit after graduation recently, so the work unit information of the third author was updated. The main corrections in the paper and the responds to the reviewer’s comments are as flowing:
Responds to the reviewer’s comments:
Reviewer #2:
- I propose to extend the description of the rolling process by the diagram, parameters in subsequent culverts, order of operations.
Response:
The rolling information has been added to Figure 2, and the main parameters and operations were extend in Chapter 2.1.2.
“The maximum rolling force and roll gat of hot mill are 4000kN and 170mm respectively. The rated power of main drive motor is 400kW. The multi-function cooling system is installed at the delivery side of the mill. The schematic diagram of the hot rolling experiments is presented in Figure 2. The slabs with a 70 mm thickness were reheated to 1200 °C in a furnace and maintained for 2 hours. Then, they were hot rolled to sheets with 4 mm thickness by multi-pass rolling. The final rolling temperature was 900 °C. Following rolling, the plates were cooled to 620 °C (industrial coiling temperature) in different cooling modes, such as air-cooling, slow-cooling with water, and fast-cooling with water. Then the plates were slowly cooled to room temperature in a pit-type furnace. Tensile testing was performed on a normal tensile sheet specimen to study the strength of different cooling processes. The thickness of the tensile specimen is 3.8 mm, and the total length is 200mm. The lengths of clamping part and body part are 50mm and 70mm, respectively. The clamping part and body part are smoothly connected with an arc with radius of 20 mm. The width of body part is 12.5 mm. The specimens cut from the hot-rolled plates were ground, polished and then etched in a 4% nitrate alcohol solution to study the microstructure with different cooling processes. The optical micrograph (OM) and scanning electron micrograph (SEM) was observed by an optical microscope(OLYMPUS BX53MRF) and scanning electron microscope(FEI Quanta 600), respectively.”
- I propose to explain in detail how subsequent layers of sheet metal were modeled in hot-rolled coil.
Response:
Thank you for your suggestion, and we have added the explanation in Chapter 2.2
During the coiling calculation of each tick step, coil with new diameters was re-modeled. By defining node data in the input file, the temperature of old layers was initialized with the previous tick step, and the temperature of new layers was initialized with coiling temperature. The coiling temperature of 620 °C.
- Has the sheet strength test been carried out? If so, propose to include the results.
Response:
The strengths of hot-rolled coil were tested before and after U-shaped cooling process application, which were presented in Figure 15. The longitudinal strength fluctuation of hot-rolled coils will cause thickness fluctuation in the subsequent cold rolling process, so the thickness fluctuation in cold rolling process was monitored and compared.
Thickness fluctuation before U-shaped cooling application
Thickness fluctuation after U-shaped cooling application
The steel company of Handan has provided only screenshots above, and forbid exporting data from the database of computer control system. Due to the quality of the screenshots, this content is not included in the paper.

Reviewer 3 Report
The aim of this paper is to show how to improve the longitudinal performance of hot rolled coils.
Except for the experimental results of the microstructures, the scientific contribution is low and not clear.
The authors already mention in the introduction that the temperature profile is expected to be low at the head and tail parts of the coil due to the high cooling rates. And fast cooling rate increases the strength since it influences the microstructure evolution. These are all know effects and in the conclusions nothing new is given.
2.1.1 Thermal simulation experiments and 2.1.2 Hot rolling experiments are describing the same process with different parameters. Why is this repetition?
It is not written how the nominal compressive stress is calculated/obtained as required by equation 1.
In 3.1 the dynamic CCT is mentioned without giving any deformation rate.
The FEM simulation results - It is not clear where the simulations results are used and why they are needed? Exact geometrical definitions such as total thickness, number of layers and specifically applied boundary conditions between layers are missing. Also no comparison is made between different cooling rates which is the main aim of this paper. The scale in Figures 8, 10 and 11 are not clear. In figures 10 and 11 the results seem to have uniaxial symmetry at the center of the coil which cannot be true. If not, it should be demonstrated how the values change over inner and outer boundary.
Conclusions are very weak and should be improved.
Many times written in the paper thermal simulation experiments, is it simulation or experiment?
I would suggest the authors to leave the FEM simulation out and focus more on the microstructural evolutions and come up with better conclusions.
Author Response
Dear Editors and Reviewers:
Thank you for your letter and comments concerning our manuscript entitled “Improvement of longitudinal performance uniformity of hot-rolled coils for cold-rolled DP980 steel” (ID: metals-740303). Those comments are all valuable and very helpful for revising and improving our paper, as well as the important guiding significance to our researches. We have studied comments carefully and have made correction which we hope meet with approval. We would like also to thank you for allowing us to resubmit a revised copy of the manuscript. In addition, the third author of this paper has changed his work unit after graduation recently, so the work unit information of the third author was updated. The main corrections in the paper and the responds to the reviewer’s comments are as flowing:
Responds to the reviewer’s comments:
Reviewer #3:
- 1.1 Thermal simulation experiments and 2.1.2 Hot rolling experiments are describing the same process with different parameters. Why is this repetition?
Response: The thermal simulation experiments was carried out to study the effect of cooling rate after coiling on the microstructure and properties, But the hot rolling experiments was carried out to study the effect of the cooling rate before coiling on the microstructure and properties. The process parameters of the thermal simulation experiments and the hot rolling experiments are different.
- It is not written how the nominal compressive stress is calculated/obtained as required by equation 1
Response: the nominal compressive stress Pc is equal to 30MPa, which had provided by reference 12.
- In 3.1 the dynamic CCT is mentioned without giving any deformation rate.
Response: The strain is 0.5 and the deformation rate is 5 s-1, which has been presented in Figure1(b).
- The FEM simulation results - It is not clear where the simulations results are used and why they are needed? Exact geometrical definitions such as total thickness, number of layers and specifically applied boundary conditions between layers are missing. Also no comparison is made between different cooling rates which is the main aim of this paper. The scale in Figures 8, 10 and 11 are not clear. In figures 10 and 11 the results seem to have uniaxial symmetry at the center of the coil which cannot be true. If not, it should be demonstrated how the values change over inner and outer boundary.
Response:
- The FEM simulation results were used to decide the parameters of U-shaped cooling, which is discussed in Line 301~314.
- Exact geometrical definitions such as total thickness, number of layer are listed in Table.2. The boundary conditions between layers were denoted by the equivalent thermal conductivity of hot-rolled coil in the radial direction, which can be calculated with equation (1).
- The aim of this paper is to discover the causes of longitudinal strength fluctuations of DP980 steel, rather than to find out an appropriate cooling rate.
- The Figures 8, 10 and 11 are improved.
- This is because a slight deviation between the observation angle and the z view, a visual shadow is formed on the 3d model. Moreover, the yellow and reddish regions are round, which indicates that the color difference does not mean the temperature difference.
- Conclusions are very weak and should be improved.
Response:
The first conclusion was corrected as following.
“The cooling rate, before coiling, has a slight influence on the microstructure and properties of hot-rolled strips. The size of ferrite and the M-A island become slightly finer with the increase of the cooling rate, before coiling, so the accelerated rolling will contribute to the increase of the strength of body and then to the improvement of the longitudinal performance uniformity of hot-rolled coils.”
In addition, a new conclusion (No.3) was added to this paper.
“The FEM simulation results show that the average cooling rate of the inner and outer layers is higher than that of the middle layers of the hot-rolled coil during and after coiling. For DP980 steel, the high cooling rate leads to the decrease of the fraction of proeutectoid ferrite and grains refinement. As a result, the strengths of the head and tail parts are higher than that of the body part of the hot-rolled strip, which is consistent with the problem of industrial production.”
- Many times written in the paper thermal simulation experiments, is it simulation or experiment?
Response: The experiment was used to discover the causes of longitudinal strength fluctuations of DP980 steel. The simulation was used to optimize the parameters of U-shaped cooling process, which is applied in industrial applications to eliminate the longitudinal strength fluctuations.

Reviewer 4 Report
The Authors focused on improvement of the longitudinal performance uniformity of hot-rolled coils. They investigated the dynamic continuous cooling transformation and the influence of the cooling mode before coiling and during coil cooling on the microstructure and mechanical properties. The research was carried out through thermal numerical simulation and hot rolling experiments. The numerical simulations have been conducted using computer program ABAQUS and the parameters of temperature cooling process of a coil were optimized and used in industrial applications. The article seems to be interesting for engineers. It creates a logical scientific research and, therefore, in my opinion could be published in "Metals" after minor revision and improvement of English language. The article is understandable but the colloquial terms are used instead of technical ones. Some of the comments on the manuscript are listed below.
- The keywords are not appropriate because they are taken directly from the title of the article.
- Line 70, 72. What do “φ450 mm x 450” and “φ8 mm x 15 mm” mean?
- Line 81. The data should be placed rather in a table than in the text.
- The basic schematic drawing of the analysed process is missing in the very beginning of the manuscript.
- Spelling mistakes (line: 77, 110, 154, 155, 160, 263, 270 and 316).
- Line 123. Instead of “hoop” should be “circumferential”.
- Line 172. Instead of “velocity acceleration” should be probably “velocity and acceleration”.
- In radial direction the resistance of thermal conductivity is in series and it is OK but in the transvers direction (circumferential) the resistance is parallel. How the equivalent resistance has been obtained?
- How the emissivity of a strip (ε) was found?
- Table 4. Could you give physical interpretation why with increasing of cooling rate (let say with stiffness increasing) the elongation increases too?
- Fig. 9. These two phases should be clearly marked on the graph: one during and the other one after coiling using for example different colours or markers. It would facilitate Readers the better understanding of these interesting phenomena.
- Fig. 10b and 11b – the legend should be added.
- The conclusions are not clearly explained and should be corrected.
Author Response
Dear Editors and Reviewers:
Thank you for your letter and comments concerning our manuscript entitled “Improvement of longitudinal performance uniformity of hot-rolled coils for cold-rolled DP980 steel” (ID: metals-740303). Those comments are all valuable and very helpful for revising and improving our paper, as well as the important guiding significance to our researches. We have studied comments carefully and have made correction which we hope meet with approval. We would like also to thank you for allowing us to resubmit a revised copy of the manuscript. In addition, the third author of this paper has changed his work unit after graduation recently, so the work unit information of the third author was updated. The main corrections in the paper and the responds to the reviewer’s comments are as flowing:
Responds to the reviewer’s comments:
Reviewer #4:
- The keywords are not appropriate because they are taken directly from the title of the article.
Response: thank you for your suggestion, we have modified keywords.
Keywords: DP980 steel; U-shaped cooling; dynamic continuous cooling transformation; microstructure and mechanical properties.
- Line 70, 72. What do “φ450 mm x 450” and “φ8 mm x 15 mm” mean?
Response: ‘‘Φ450 mm x 450’’ means the diameter of work roll is 450mm, roll body length is 450mm; the diameter of cylindrical samples is 8mm, the length is 15mm.
- Line 81. The data should be placed rather in a table than in the text.
Response: thank you for your suggestion, we have listed the data in Fig. 1, and deleted them from text. (Chapter 2.1.1)
- The basic schematic drawing of the analysed process is missing in the very beginning of the manuscript.
Response: We have already stated the analysed process at the end of the introduction, so we did not make a schematic drawing to explain the research and analysis process.
- Spelling mistakes (line: 77, 110, 154, 155, 160, 263, 270 and 316).
Response: I am very sorry for our mistake, and we have modified them, which were listed as follows:
Line: 77 900 oC → 900 °C
Line: 110 The coiling stage and coil discharged stage → The coiling stage and the coil discharged stage
Line: 154 W/(m2 K) → W/(m2 K)
Line: 155 W/(m2 K4) → W/(m2 K4)
Line: 160 The dynamic CCT of the DP980 steel curve → The dynamic CCT curve of the DP980 steel
Line: 263 ΔTH, LH1 and LH2 → ΔTH, LH1 and LH2
Line: 270 ΔTT, LH1 and LH2 → ΔTT, LT1 and LT2
Line: 316 assistance → assistances
- Line 123. Instead of “hoop” should be “circumferential”.
Response: We have modified it according to your suggestion. “hoop” →“circumferential”
- Line 172. Instead of “velocity acceleration” should be probably “velocity and acceleration”.
Response: thank you for your suggestion, and we have modified “velocity acceleration” into ‘’accelerated rolling’’ to make it clear.
- In radial direction the resistance of thermal conductivity is in series and it is OK but in the transvers direction (circumferential) the resistance is parallel. How the equivalent resistance has been obtained?
Response: In the transvers and circumferential direction, the resistance of thermal conductivity is decided by the thermal conductivity of steel. In addition, the hot-rolled coil was simplified to a combination of multilayer rings, and each ring had the same temperature, so the thermal conductivity in the circumferential can be ignored.
- How the emissivity of a strip (ε) was found?
Response: the emissivity of a strip is 0.8, which was defined according to reference [12], and it are reflected in the manuscript. (Line: 179)
- Table 4. Could you give physical interpretation why with increasing of cooling rate (let say with stiffness increasing) the elongation increases too?
Response:
As the cooling rate increases, the grains of the strip become finer. According to the theory of metallization, grain refinement is beneficial to both strength and plasticity.
- Fig. 9. These two phases should be clearly marked on the graph: one during and the other one after coiling using for example different colours or markers. It would facilitate Readers the better understanding of these interesting phenomena.
Response: The finishing time of coil discharging is added to figure 9.
- Fig. 10b and 11b – the legend should be added.
Response: The legends of Fig. 10b and 11b were presented in the below of temperature curve.
- The conclusions are not clearly explained and should be corrected.
Response: The first conclusion was corrected as following.
“The cooling rate, before coiling, has a slight influence on the microstructure and properties of hot-rolled strips. The size of ferrite and the M-A island become slightly finer with the increase of the cooling rate, before coiling, so the accelerated rolling will contribute to the increase of the strength of body and then to the improvement of the longitudinal performance uniformity of hot-rolled coils.”
In addition, a new conclusion (No.3) was added to this paper.
“The FEM simulation results show that the average cooling rate of the inner and outer layers is higher than that of the middle layers of the hot-rolled coil during and after coiling. For DP980 steel, the high cooling rate leads to the decrease of the fraction of proeutectoid ferrite and grains refinement. As a result, the strengths of the head and tail parts are higher than that of the body part of the hot-rolled strip, which is consistent with the problem of industrial production.”

Round 2
Reviewer 3 Report
I agree with most of the responses of the authors. There has been a significant improvement of the paper. I can only suggest authors to include a general comment on ΔTH and ΔTT values in conclusions even though this paper is specific to one type of steel.